# Glycolytically impaired Drosophila glial cells fuel neural metabolism via β-oxidation

Ellen McMullen[1,4], Helen Hertenstein[2,4], Katrin Strassburger[2], Leon Deharde[2], Marko Brankatschk [3] ✉ & Stefanie Schirmeier [2] ✉

Neuronal function is highly energy demanding and thus requires efficient and constant metabolite delivery by glia. Drosophila glia are highly glycolytic and provide lactate to fuel neuronal metabolism. Flies are able to survive for several weeks in the absence of glial glycolysis. Here, we study how Drosophila glial cells maintain sufficient nutrient supply to neurons under conditions of impaired glycolysis. We show that glycolytically impaired glia rely on mitochondrial fatty acid breakdown and ketone body production to nourish neurons, suggesting that ketone bodies serve as an alternate neuronal fuel to prevent neurodegeneration. We show that in times of long-term starvation, glial degradation of absorbed fatty acids is essential to ensure survival of the fly. Further, we show that Drosophila glial cells act as a metabolic sensor and can induce mobilization of peripheral lipid stores to preserve brain metabolic homeostasis. Our study gives evidence of the importance of glial fatty acid degradation for brain function, and survival, under adverse conditions in Drosophila.

The nervous system consumes a disproportionally large amount of energy compared to its size[1]. Carbohydrates are the preferred energy source and are imported from circulation at large quantities[2]. Glucose is metabolized glycolytically. The highest glycolytic activity has been associated with glial cells, which produce lactate to fuel neuronal oxidative metabolism[3–6]. This metabolic coupling of glial cells and neurons has been termed the "Astrocyte-Neuron Lactate Shuttle" (ANLS) and has been shown to be conserved across species[7,8]. Under optimal conditions glial cells are able to conserve energy by producing glycogen[9] that can be used to fuel peaks of neuronal energy demand, or bridge short periods of malnutrition[9,10]. But is glial glycogen the only alternative fuel used in periods of glucose deprivation?

In the periphery, ketone bodies and fatty acids (FA) are used to produce the energy required to endure malnutrition. The use of FA by the brain to gain energy has been debated[11–14]; even though major rate-limiting enzymes of β-oxidation are expressed in the nervous system[15,16]. It has been suggested that glial cells are capable of performing mitochondrial FA degradation under certain conditions[12,17,18].

Here, we present evidence for the importance of glial mitochondrial ß-oxidation for neuronal function and animal survival in Drosophila. Upon glycolytic impairment or starvation-induced hypoglycemia, genes predicted to function in glial ß-oxidation and ketogenesis become essential to support neuronal function and prevent neurodegeneration. Further, we demonstrate that dysregulation of glial carbohydrate or lipid metabolism induces lipid mobilization from peripheral storage organs. This metabolic interorgan communication is likely mediated by glial lipoprotein GLaz.

## Results

### Glial cells can switch to β-oxidation
Glial glycolysis is essential for neuronal survival in Drosophila, while neuronal glycolysis is dispensable[8]. Interestingly, glial glycolytic knockdown only reduces lifespan by about half[8]. Therefore, we speculated that glia use FA not only to sustain very specific neuronal function, such as memory formation, as has been suggested previously[14], but as a general alternative fuel to ensure global brain

[1]Department of Molecular Biology and Genetics, University of South Bohemia, České Budějovice, Czech Republic. [2]Zoology and Animal Physiology, Faculty of Biology, Technische Universität Dresden, Dresden, Germany. [3]Biotechnologisches Zentrum, Technische Universität Dresden, Dresden, Germany. [4]These authors contributed equally: Ellen McMullen, Helen Hertenstein. ✉e-mail: marko.brankatschk@tu-dresden.de; stefanie.schirmeier@tu-dresden.de

function and prolong survival. To analyze glial or neuronal breakdown of FA via β-oxidation, we targeted the enzymes Carnitine palmitoyl-transferase 2 (CPT2, CG2107) and Mitochondrial trifunctional protein α subunit (Mtpα, CG4389). CPT2 imports acyl-carnitines into the mitochondria, while Mtpα possesses Enoyl-CoA hydratase and a Hydroxyacyl-CoA dehydrogenase activity and catalyzes the breakdown of Acyl-CoA in the mitochondria. Knockdown of β-oxidation alone in neurons or glia of adult animals did not reduce lifespan (Fig. 1a, S1A, *CPT2[i]*, *Cherry[i]* and *Mtpα[i]*, *Cherry[i]*). While neuronal metabolic manipulations did not cause any abnormal phenotype (Fig. S1A), simultaneous reduction of β-oxidation and pyruvate kinase (Pyk, a key enzyme of glycolysis) in adult glial cells significantly reduced lifespan compared to glycolysis knockdown alone (Fig. 1a), indicating that glial β-oxidation is essential upon glycolytic impairment. In order to assess whether inhibition of glial glycolysis indeed affects the rate of β-oxidation, we analyzed mitochondrial morphology (Fig. S2A). It was previously shown that the degree of mitochondrial fusion or fission reflects the metabolic state[19]. Mitochondrial β-oxidation is more efficient when the mitochondria are in a highly fused state[20]. Indeed, mitochondria in glycolytically impaired glial cells show a much higher degree of fusion than in glial cells of control animals (Fig. S2A, B), possibly allowing a glial switch to β-oxidation.

## Starvation induces a metabolic switch

Under starvation conditions, carbohydrate supply in the animal is limited[21] and alternative substrates must be used for glial energy production. To examine the effect of carbohydrate restriction, we starved adult flies and assessed their lifespan. Animals with glycolytically impaired glia (*Pyk[i]*,*Cherry[i]*) succumbed to starvation at the same rate as controls (Fig. 1b), indicating that starving animals do not rely on carbohydrate metabolism. In contrast, flies with glial β-oxidation knockdown (*CPT2[i]*,*Cherry[i]* or *Mtpα[i]*,*Cherry[i]*) showed increased starvation susceptibility (Fig. 1b). The lifespan of those animals is strongly reduced (by 30%). Loss of glial glycolysis in addition to β-oxidation (*Pyk[i]*,*CPT2[i]* or *Pyk[i]*, *Mtpα[i]*) did not enhance the phenotype (Fig. 1b). Thus, also under physiological conditions, like nutrient restriction, glia seem to use β-oxidation to ensure neuronal energy supply.

## Loss of β-oxidation boosts degeneration

Since neurodegeneration is most likely the cause of premature death when glial glycolysis is impaired[8], we speculated this may be accelerated upon additional loss of β-oxidation. To test this, we assessed the activity of the animals, since neurodegeneration often correlates with a loss in coordination and mobility[22]. *CPT2* or *Mtpα* knockdown

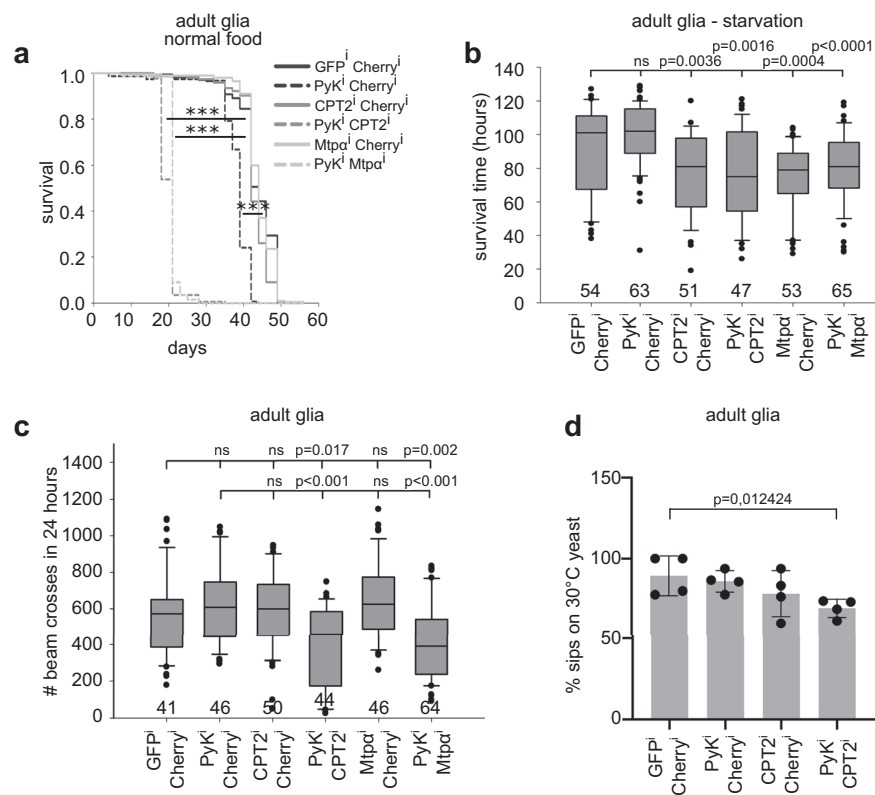

**Fig. 1 | Glia rely on β-oxidation upon loss of glycolysis or nutrient restriction.** **a** Lifespan of glial glycolysis/β-oxidation knockdown animals (CPT2, Pyk and Mtpα, Pyk) is significantly reduced compared to glycolysis only knockdown (Pyk, Cherry). For statistical analysis log rank test was used. Multiple Comparison was performed using the Holm-Sidak method. ***: *p* < 0,000001. Three independent experiments were performed. Total number of animals: GFP[i],Cherry[i]: 174, Pyk[i],Cherry[i]: 154, CPT2[i],Cherry[i]: 154, Pyk[i], CPT2[i]: 171, MTP[i],Cherry[i]: 200, Pyk[i], MTP[i]: 198. **b** In contrast to glial glycolysis loss (Pyk, Cherry), glial β-oxidation knockdown (CPT2, Cherry and Mtpα, Cherry) induces starvation susceptibility. Numbers below each bar indicate number of animals used. Three independent experiments were performed. Statistically significant differences were determined using one-way ANOVA analysis. Box plots: the boundary of the box indicate the 25th and 75th percentile, the line within the box marks the median. Whiskers (error bars) above and below the box indicate the 90th and 10th percentiles. Dots indicate outliers. **c** Locomotive

activity is reduced in glial glycolysis/β-oxidation knockdown animals (CPT2, Pyk and Mtpα, Pyk) at the age of one week. Numbers below each bar indicate number of animals used. Three independent experiments were performed. Statistically significant difference in activity was determined using two-tailed Mann-Whitney rank sum test. Box plots: the boundary of the box indicate the 25th and 75th percentile, the line within the box marks the median. Whiskers (error bars) above and below the box indicate the 90th and 10th percentiles. Dots indicate outliers. **d** Glial glycolysis, β-oxidation knockdown animals lose their ability to choose yeast grown at 30 °C over yeast grown at 10 °C as food source. *N* = 4 independent experiments, *n* = 48 animals. Significant differences in numbers of feeding events compared to control animals were determined using one-tailed student's t-test. Graphs: Mean ± standard deviation is shown. Dots represent individual values. Source data are provided as a Source Data file.

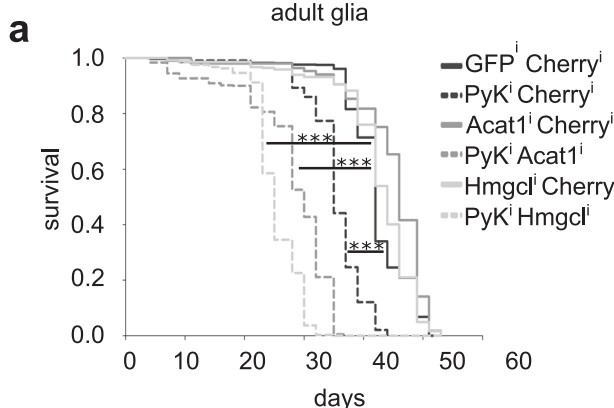

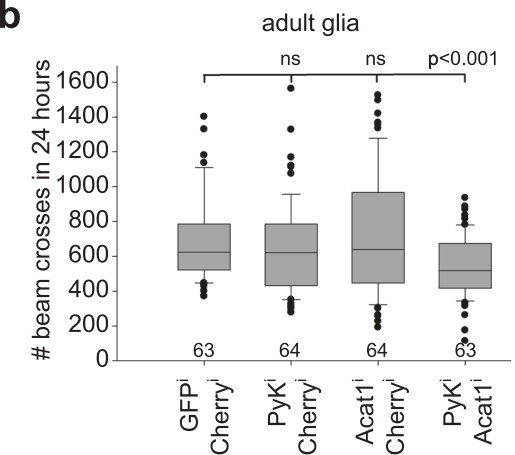

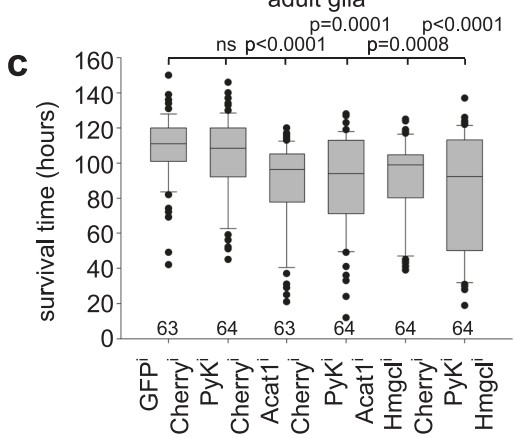

**Fig. 2 | Glycolytically impaired glial cells produce ketone bodies via mitochondrial β-oxidation. a** Simultaneous knockdown of ketone body synthesis (knockdown of Acat1 or Hmgcl) and glycolysis (Pyk) in adult glia significantly reduces lifespan compared to glycolysis only knockdowns. Ketone body synthesis only knockdown animals show wild-typic lifespan. For statistical analysis log rank test was used. Multiple Comparison was performed using the Holm-Sidak method. ***: $p < 0,000001$. Two to five independent experiments were performed. Total number of animals: GFP[i],Cherry[i]: 647, Pyk[i],Cherry[i]: 580, Acat1[i],Cherry[i]: 560, Pyk[i], Acat1[i]: 563, Hmgcl[i],Cherry[i]: 367, Pyk[i], Hmgcl[i]: 376. **b** Average locomotive activity is significantly decreased in one week old Pyk, Acat1 double knockdown animals. Numbers below each bar indicate number of animals used. Three independent experiments were performed. Statistically significant difference in activity was determined using two-tailed Mann-Whitney rank sum test. Box plots: the boundary of the box indicate the 25th and 75th percentile, the line within the box marks the median. Whiskers (error bars) above and below the box indicate the 90th and 10th percentiles. Dots indicate outliers. **c** As glial loss of β-oxidation, glial loss of ketone body synthesis induces starvation susceptibility. Numbers below each bar indicate number of animals used. Two independent experiments were performed. Statistically significant differences were determined using one-way ANOVA analysis. Box plots: the boundary of the box indicate the 25th and 75th percentile, the line within the box marks the median. Whiskers (error bars) above and below the box indicate the 90th and 10th percentiles. Dots indicate outliers. Source data are provided as a Source Data file.

To confirm that glial loss of both, glycolysis and β-oxidation, induces neurodegeneration, we directly analyzed brain morphology using semi-thin head sections (Fig. S3B–K). Neurodegenerative brain areas show abnormal tissue organization[8,24]. Indeed, we detected holes in the cortical regions of glial double knockdown brains, already at two weeks of age (Fig. S3E, G), while impaired glial glycolysis alone induced signs of neurodegeneration only later (38 days, compare Fig. S3I[8]). Inhibition of glial β-oxidation alone, or respective neuronal manipulations, did not lead to neurodegeneration (Fig. S1D–I, S3). Further, we analyzed whether glial metabolic dysfunction also affects neuronal function on the molecular level. Indeed, a presynaptic active zone component, Bruchpilot (Brp), is deregulated upon glial glycolysis and β-oxidation knockdown (Fig. S3L). Tight regulation of active zone components has been previously shown to be essential for the correct output strength and proper function of neuronal circuits[25–28]. In conclusion, loss of glial glycolysis causes premature loss of neuronal function and neurodegeneration[8] which is strongly accelerated when glial β-oxidation is lost concomitantly. Thus, glial FA degradation is sufficient to support neuronal survival and function for some time.

### Glial cells provide ketone bodies

Neurons can metabolize ketone bodies derived from circulation[29,30]. Ketone bodies are generated via β-oxidation in the liver, e.g. upon carbohydrate restriction or hunger. We speculated that dietary restriction could force glia to produce ketone bodies from β-oxidation to fuel neurons. To test this, we performed glia-specific knockdown of either Acetyl-CoA C-acetyltransferase (Acat1, CG10932, orthologue of human and mouse ACAT1), required to convert Acetyl-CoA into Acetoacetyl-CoA, or 3-Hydroxymethyl-3-methylglutaryl-CoA lyase (Hmgcl, CG10399, orthologue of human and mouse HMGCL), catalyzing the cleavage of 3-Hydroxy-3-methylglutaryl-CoA to generate acetoacetate and Acetyl-CoA. The lifespan of flies with impaired glial ketogenesis and glycolysis is reduced to a level comparable to β-oxidation, glycolysis knockdowns (Fig. 2a, 1a). To examine if progressive neurodegeneration is responsible for the early lethality of glial glycolysis, ketogenesis double knockdown animals, we assayed their mobility. We found that one-week-old animals with inhibited glial ketogenesis and glycolysis are less active, phenocopying double β-oxidation, glycolysis knockdown animals (Figs. 2b, 1c). In addition, glial knockdown of ketone body synthesis induces starvation susceptibility as seen in glial β-oxidation knockdowns (Fig. 2c, 1b). Taken together

flies showed control activity (Fig. 1c, S3A). In contrast, we found that glial Pyk[i] reduced the activity of aged animals (22 days), indicating progressive decline of neuronal activity (Fig. S3A). This phenotype occurs earlier when glial β-oxidation is knocked down in addition. Both, Pyk[i],CPT2[i] and Pyk[i], Mtpα[i] double knockdown animals exhibited decreased mobility already after one week (Fig. 1c) and died at the age of three weeks (Fig. 1a). In contrast, respective neuronal metabolic manipulations did not result in abnormal phenotypes (Fig. S1B, C). A more specific read-out for neuronal functionality is an odor-based food choice assay where flies distinguish between two different yeast-based food sources based on olfactory and gustatory cues[23]. Indeed, double knockdown of glial glycolysis and β-oxidation causes animals to lose their ability to distinguish between different food types (Fig. 1d).

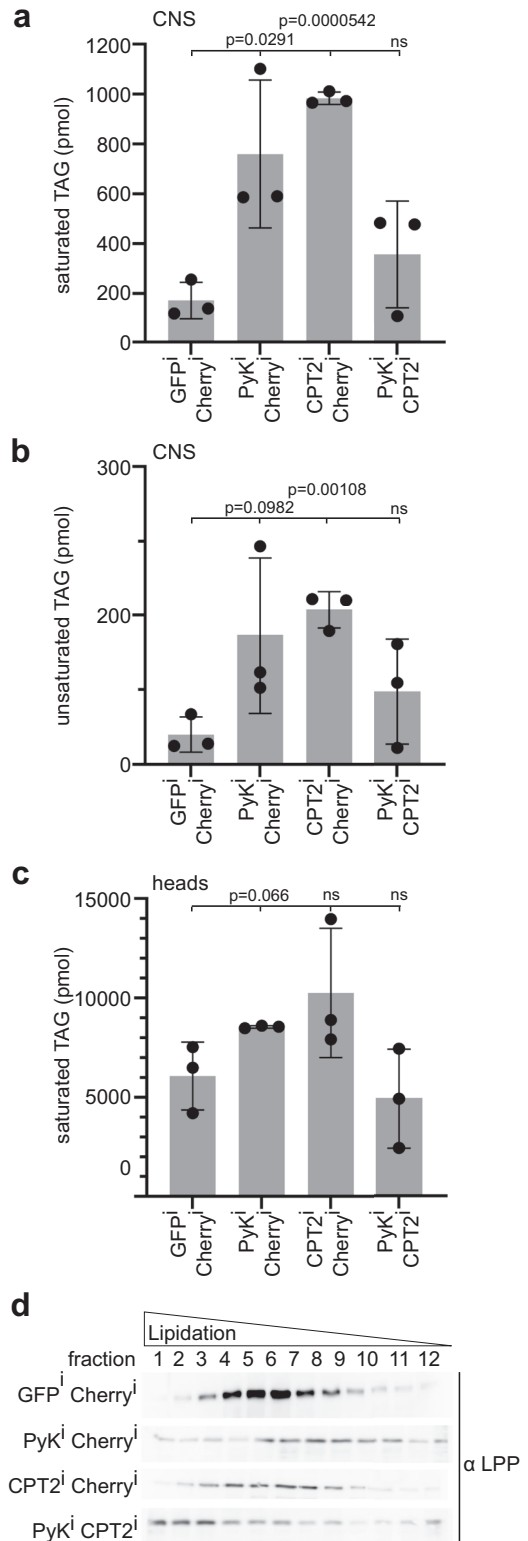

**Fig. 3 | Impaired glia metabolism affects lipid homeostasis. a, b** Levels of saturated (**a**) and unsaturated (**b**) TAGs in the CNS are significantly increased upon glial glycolysis (Pyk) or β-oxidation (CPT2) knockdown. $N = 3$ independent experiments with a total of 15 animals ($n$). To assess significant differences between the genotypes two-tailed Mann-Whitney rank sum test was used. Graphs: means ± standard deviation; dots are values of individual measurements. **c** Yields of saturated TAGs in total head samples are significantly increased upon glial glycolytic impairment. $N = 3$ independent experiments with a total of 15 animals ($n$). To assess significant differences between the genotypes two-tailed Mann-Whitney rank sum test was used. Graphs: means ± standard deviation; dots are values of individual measurements. **d** Immunoblot depicting the Lpp signal distribution in different fractions of a density gradient. N2-3, $n = 30$ flies per experiment Fraction 1 (left) represents sample with lowest density and Fraction 12 (right) the sample with the highest density. Source data are provided as a Source Data file.

increased TAG levels in the CNS (Fig. 3a, b). Interestingly, glial knockdown of glycolysis produced similar results indicating a complex regulation of TAG levels in the brain. Such a regulation could involve the fat body (a Drosophila tissue analogous to mammalian liver and white adipose tissue[31]), as reduced glial metabolite turnover was shown to initiate a feedback to the periphery[32]. The brain is surrounded by fat body in the head and thin-layer chromatography experiments using brain and head samples showed that the TAG content of brains is negligible compared to the TAG content of heads (Fig. S4A). Therefore, we analyzed head samples, in order to test whether changes in glial metabolism affect TAG levels in fat body cells. MS data reveal similar changes in saturated TAG levels in the head fat body as seen in brain samples (compare Fig. 3a to c), indicating that metabolic deficiency of the brain triggers compensatory mechanisms in the periphery.

In this case, lipids should be mobilized and enter circulation. In Drosophila, Lipophorin particles (Lpp) are the main systemic lipid carriers[33,34]. To assess the lipid load of Lpp, we separated them in a density gradient. High lipid loads (i.e. high diacylglycerol (DAG) loads) force lipoprotein particles to accumulate in less dense fractions of the gradient[33,35]. We found that lipoprotein particles are more lipidated in glial glycolysis, β-oxidation double knockdown animals (Fig. 3d) indicating an activation of systemic lipid traffic. However, this experiment reports the total lipid load of LPP, but not the general amount of these carriers in circulation. LPP have a shell formed by phospholipids (PL)[33]. In order to measure the amount of LPPs in addition to their degree of lipidation, we used MS to quantify the amount of DAG and PL present in hemolymph samples. Indeed, we found that DAG and PL levels differ between genotypes. Especially glial double glycolysis, β-oxidation knockdown animals show much lower PL levels (Fig. S4D), while DAG levels are the same or even higher than in control animals (Fig. S4C). Thus, the DAG/PL ration, indicating Lpp lipidation, is much higher in glial double knockdown animals compared to control animals (Fig. S4C–E). Since the fat body represents the only LPP source, our data suggests the regulation of fat body cells by glia. Taken together, our results show that glial metabolic insufficiency affects TAG levels in the fat body and the mobilization of storage lipids. Further, our data suggest an interorgan communication between glia and the periphery.

### Glial GLaz loss mimics glial glycolysis & β-oxidation loss
Direct communication of glia with peripheral lipid stores requires a secreted messenger. Deregulation of this signal molecule in glial cells should cause similar phenotypes as observed in animals with impaired glial glycolysis and/or β-oxidation. Therefore, we tested glial knockdowns of three candidate genes for starvation susceptibility: Drosophila insulin-like peptide 6 (*dIlp6*), secreted decoy of Insulin receptor (*Sdr*) and Glial Lazarillo (*GLaz*), the Drosophila homologue of the apolipoprotein ApoD. Glial knockdown of *dIlp6* or *Sdr* did not influence starvation resistance. In contrast, glial knockdown of GLaz induced strong starvation susceptibility, phenocopying loss of β-

this data suggests that upon loss of glial glycolysis or upon starvation glial cells produce ketone bodies to fuel neuronal metabolism.

### Glia-fat body interorgan crosstalk mobilizes energy stores
β-oxidation is the breakdown of fatty acids, which are mostly stored as triacylglycerides (TAGs) in lipid droplets. To test whether glial metabolic impairment changes TAG yields in brains, we measured TAG levels in lipid extracts from adult brains by mass spectrometry (MS, Fig. 3a, b). As expected[12], inhibited glial β-oxidation resulted in

oxidation (Fig. 4a). GLaz mutants were previously reported to be starvation susceptible[36]. Our data indicate that this phenotype is caused by the loss of GLaz in the glial cells. To further analyze the effect of glial loss of GLaz, we performed lifespan experiments (Fig. 4b). Glial knockdown of GLaz results in a severe reduction in lifespan, a similar phenotype as seen in glial glycolysis, β-oxidation double knockdowns. We further analyzed if glial GLaz knockdown animals also phenocopy the more specific phenotypes we see in glial glycolysis, β-oxidation double knockdown animals, such as behavioral defects and changes in the levels of the active zone component Brp. Indeed, animals that lack glial GLaz are unable to choose between two food sources based on their odor at 14d of age (Fig. 4c). Furthermore, strong accumulation of Brp occurs in the neurons of these animals as is the case in neurons of glial glycolysis, β-oxidation double knockdown animals (Fig. 4d). It has previously been reported that GLaz mutants show deregulation of peripheral lipid homeostasis and lipid storage in the fat body[36]. In sum, loss of glial Glaz results in many phenotypes reminiscent of deregulated glial metabolism. Therefore, secreted GLaz, or GLaz containing Lipoprotein particles, is a compelling candidate to convey a metabolic signal from the glial cells to the fat body. However, further experiments will be needed to unravel the molecular interactions underlying signal generation and perception.

## Discussion

Neurons need a large amount of energy to function. This energy is provided with the help of glial cells that are glycolytically active, providing neurons with metabolites such as lactate[5,7,8]. Here, we show that, in Drosophila, glial metabolism is not limited to glycolysis but is flexible and can switch to the use of FA to fuel neuronal metabolism when carbohydrate metabolism is insufficient.

It has long been thought that the brain is mostly restricted to carbohydrates as an energy source[16]. Long-term starvation reduces sugar availability. We show that under such circumstances Drosophila glial cells degrade FAs and provide ketone bodies to prevent neuronal degeneration. Further, animals with a glial loss of glycolysis and β-oxidation or ketogenesis are more susceptible to starvation. Thus, we propose that glial β-oxidation is essential to maintain brain energy homeostasis when carbohydrate metabolism is restricted genetically or due to hypoglycemia.

In the insect CNS, lipid droplets are found in glial cells[12]. These lipid droplets are most likely initially used to fuel β-oxidation, since loss of β-oxidation leads to a glial accumulation of lipid droplets[12]. In mammals no lipid droplets are found in glial cells, but oligodendrocytes produce large amounts of lipid-rich myelin. The primary function of myelin is thought to be insulation of the axon, to allow fast signal conduction. Interestingly, Asadollahi et al. show that myelin can also be used as energy storage under conditions of carbohydrate restriction[37]. Here, mouse oligodendrocytes use β-oxidation to maintain energy homeostasis in the optic nerve at the expense of myelin thickness. Also, oligodendrocyte-specific knockout of *Glut1*, limiting glucose uptake, leads to a reduction in myelin thickness in vivo[37]. In addition, MRI scans of patients suffering from anorexia nervosa show a loss of white matter, indicating that severe nutrient restriction in humans can also cause oligodendrocytes to metabolize lipids stored in the myelin sheath to maintain cerebral energy homeostasis[38,39]. This indicates that the glial ability to use β-oxidation is conserved from flies to humans.

In addition to using glia-intrinsic lipid stores, Drosophila glial cells can signal to the periphery to mobilize organismal energy stores to maintain adequate energy supply to neurons. This indicates a new route of interorgan communication, used to maintain energy homeostasis in the brain. We propose that glia-derived GLaz, the Drosophila ApoD homolog, is perceived by the fat body, and that GLaz levels confer information about the metabolic state of the nervous system. This interorgan communication is likely to be conserved, since loss of cerebral β-oxidation in mammals also induces changes in peripheral metabolism[13]. Mammalian ApoD, as Drosophila GLaz, is mainly expressed by glial cells: astrocytes, oligodendrocytes and their precursors[40]. Interestingly, ApoD levels are deregulated in myelin sheath disorders[41].

The metabolic flexibility of glial cells to switch to β-oxidation is likely essential for maintaining energy homeostasis under restrictive conditions from Drosophila to humans. A better knowledge of the metabolic plasticity of the nervous system will likely prove important for understanding the effects of malnutrition and metabolic deregulation, a symptom of several neurodegenerative diseases. Since Drosophila glial cells are a metabolic sensor regulating systemic lipid mobilization—a feature that might be conserved—such deregulation could in addition have systemic effects. In the future, knowledge of the underlying interorgan communication will help to devise treatments for diverse diseases in which metabolic deregulation occurs.

## Methods

### Fly stocks

Flies were maintained at room temperature on standard food. Fly crosses were kept at 18 °C for 4 weeks on standard food. After hatching, mated female progeny of the correct genotype were transferred to 29 °C and flipped onto new food every second day.

*repoGal80*[42], *tubGal80ts*[43], *elavGal4*[44], *repoGal4*[45,46], lacZ[47]; *tub-Gal4, tub-Gal80ts* stock was a kind gift from Aurelio Teleman. Flies obtained from Bloomington Drosophila stock center: *elavGal4* (BL8765, BL8760), *tubGal80ts* (BL7017), *UAS-mito-GFP* (BL8443), Cherry[i] (BL35785), GFP[i] (BL9331), *CPT2*[HMJ23935] (BL62455), *Acat1*[HMC03340] (BL51785), *Hmgcl*[HMC03435] (BL5186); Flies obtained from VDRC: *PyK*[GD12123], *MTPα*[GD1129], *Glaz*[GD4806]

### RNA interference strategy

Using the Gal4 UAS Gal80ts system allows restricting dsRNA expression to a specific cell type in the adult animal. As controls mock-dsRNAs were used directed against GFP or Cherry to trigger the RNA interference machinery without inducing any knockdown. When expressing different numbers of targeting dsRNAs, the number of transgenes was equalized by co-expressing non-targeting mock-dsRNAs to make sure the amount of Gal4 is not limiting for knockdown efficiency. As wild type control e.g. double knockdown of GFP and Cherry was used. The driver lines used here, are the glia-specific driver line repo-Gal4[45,48,49] and the neuron-specific driver line elav-Gal4[44,50,51]. Knockdown efficiency of the different dsRNA lines is shown in Figure S5. The dsRNA line used for Acat1 (*Acat1*[HMC03340]) has been validated previously[14].

### Lifespan analysis

Mated females were collected in vials containing 20 flies and kept at 29 °C for the duration of the experiment. Vials were flipped three times per week onto fresh food, deaths were recorded, and survival rates were calculated using log-rank test. For statistical analysis log rank test was used. Multiple Comparison was performed using the Holm-Sidak method. N indicates the number of independent experiments, while n indicates the number of individual animals. Raw data for lifespan experiments can be found in supplementary data file 1.

### Starvation assay

Mated females were raised on standard food for one week at 29 °C, then transferred to individual capillaries containing plain agar. Capillaries were placed in a Drosophila activity monitor with a 12-hour light-dark cycle at 29 °C. Flies were counted as dead once movement ceased completely (and did not resume anymore until the end of the experiment). The animals were monitored for 7 days. Statistically significant differences were determined using one-way ANOVA analysis. N indicates the number of independent experiments, while n indicates the

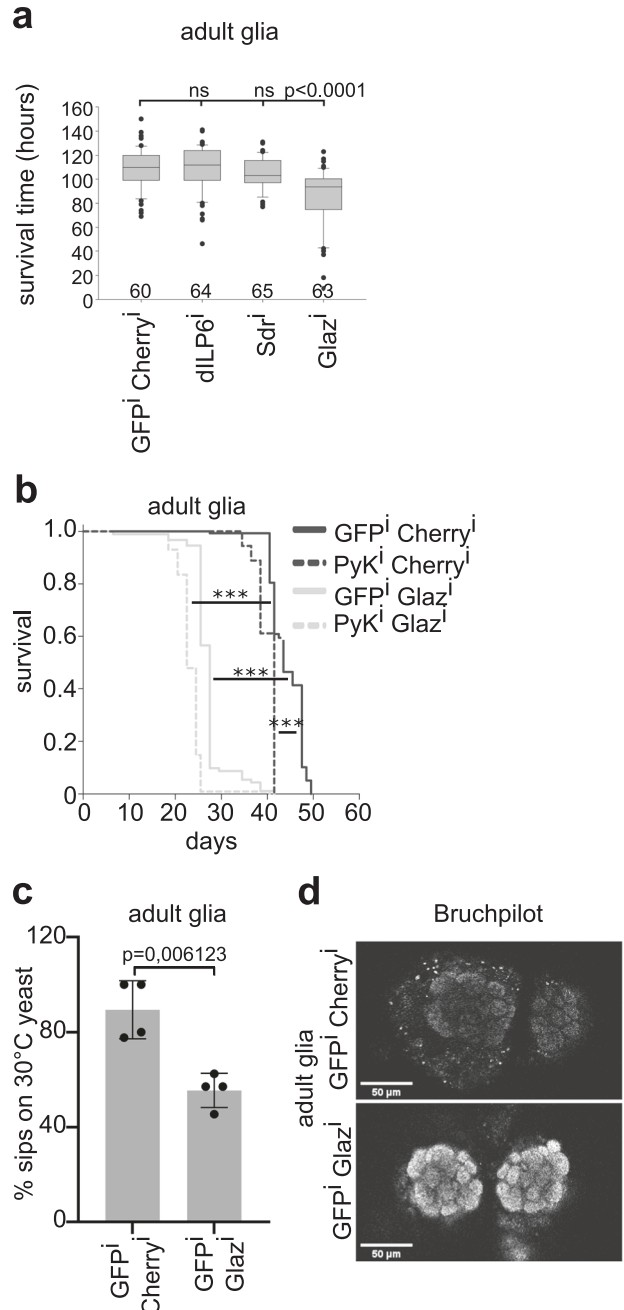

**a** adult glia

**b** adult glia

**c** adult glia

**d** Bruchpilot

**Fig. 4 | Glial loss of GLaz phenocopies glial glycolysis, β-oxidation double knockdown. a** In contrast to glial loss of dllp6 or Sdr, glial loss of GLaz induces starvation susceptibility. Numbers below each bar indicate number of animals used. Two independent experiments were performed. Statistically significant differences were determined using one-way ANOVA analysis. Box plots: the boundary of the box indicate the 25th and 75th percentile, the line within the box marks the median. Whiskers (error bars) above and below the box indicate the 90th and 10th percentiles. Dots indicate outliers. **b** Glial GLaz knockdown animals show reduced lifespan. For statistical analysis log rank test was used. Multiple Comparison was performed using the Holm-Sidak method. ***: $p < 0,000001$. Two independent experiments were performed. Total number of animals: GFP$^i$,Cherry$^i$: 138, Pyk$^i$,Cherry$^i$: 18, GFP$^i$,Glaz$^i$: 92, Pyk$^i$, Glaz$^i$: 115. **c** In contrast to control animals, glial *GLaz* knockdown animals cannot distinguish yeast grown at 30 °C from yeast grown at 10 °C at the age of 14 days. $N = 4$ independent experiments, $n = 48$ animals. Significant differences in numbers of feeding events compared to control animals were determined using one-tailed student's t-test. Graphs: Mean ± standard deviation is shown. Dots represent individual values. **d** Adult brains of 17-day-old glial knockdown animals where stained for the active zone component Brp. Brp accumulates in neurons of glial *GLaz* knockdown animals. $N = 2$ independent experiments, $n = 5$ animals. Source data are provided as a Source Data file.

### Semi-thin epon sections

Heads of adult flies, aged for 17 or 38 days at 29 °C, were embedded in Epon as described previously[48]. 1 μm semi-thin sections were cut using a EM UC7 microtome (Leica), stained with toluidine blue and imaged using a Zeiss Axiophot. At least 6 animals from 3 independent experiments were sectioned.

### Analysis of mitochondrial shape

Brains of 17-day-old animals raised at 29 °C were dissected and fixed in 4% paraformaldehyde in PBS for 1.5 h at RT. Images were acquired using a Leica TCS SP8 confocal laser scanning microscope, using the objective HC PL APO CS2 40x/1.30 OIL, numerical aperture 1.3, pinhole 999.61mAU, 600 Hz, 2048×2048 resolution and z slice distance of 0.3 μm. 30 μm stacks were scanned. Images were 3D reconstructed using Bitplane Imaris. An area of 512×512 pixels was selected, reconstruction was carried out for 13,5 μm stacks starting at 7,5 μm and ending at 21 μm of respective 30 μm stacks. Parameters used: smoothening active and set to 0.189, local background subtraction active and set to 0.5 μm, threshold set to 4. The volume data (total volume and volume units) was extracted and quantified. Wilcoxon rank-sum test was used for statistical analysis. N indicates the number of independent experiments, while n indicates the number of individual animals. Box plots: the boundary of the box indicate the 25th and 75th percentile, the line within the box marks the median. Whiskers (error bars) above and below the box indicate the 90th and 10th percentiles. Dots indicate outliers.

### Hemolymph extraction

Thirty flies were kept for 16 days on 29 °C before being frozen at −20 °C for at least 4 h in TLC-buffer. After thawing hemolymph was extracted from flies by centrifugation (500 g, RT, 1 min).

### Isopycnic gradient

An isopycnic gradient was achieved by centrifuging hemolymph samples in 0.5 g/ml KBr solution for 16 h at 4 °C, 15,000 g in vacuum. 12 fractions were taken from each sample and proteins were precipitated by the methanol/chloroform technique[52]. Fractions were analyzed for Lpp-content by Immunoblot using anti-Lpp[53] as described previously[33].

### Mass spectrometry analysis of lipidome

Mass spectrometry-based lipid analysis was performed from CNS samples, head samples or hemolymph samples by Lipotype GmbH (Dresden, Germany) using Lipotype Shotgun Lipidomics technology[54,55]. 5 CNS, 5 fly heads or 30 flies (all after 16 days at 29 °C)

number of individual animals. Box plots: the boundary of the box indicate the 25th and 75th percentile, the line within the box marks the median. Whiskers (error bars) above and below the box indicate the 90th and 10th percentiles. Dots indicate outliers.

### Drosophila activity monitoring (DAM)

Mated females aged for either one or three weeks at 29 °C on standard food were loaded into individual capillaries containing 5 % sucrose in agar. Capillaries were placed in a Drosophila activity monitor with a 12-hour light dark cycle at 29 °C. The activity of the flies over a 24-hour period was recorded and statistically significant difference in activity was determined using two-tailed Mann-Whitney rank sum test. N indicates the number of independent experiments, while n indicates the number of individual animals. Box plots: the boundary of the box indicate the 25th and 75th percentile, the line within the box marks the median. Whiskers (error bars) above and below the box indicate the 90th and 10th percentiles. Dots indicate outliers.

were collected and put into ice cold SLB-buffer (Supported Lipid Bilayer Buffer: 20 mM Tris pH 8.0, 300 mM KCl, 1 mM $MgCl_2$) and frozen at −80 °C or −20 °C. Lipids were extracted by the two-step chloroform/methanol procedure[54]. Samples were spiked with internal lipid standard mixture containing: CDP-DAG 17:0/18:1, ceramide 18:1;2/17:0 (Cer), diacylglycerol 17:0/17:0 (DAG), lyso-phosphatidate 17:0 (LPA), lyso-phosphatidylcholine 12:0 (LPC), lyso-phosphatidylethanolamine 17:1 (LPE), lyso-phosphatidylinositol 17:1 (LPI), lyso-phosphatidylserine 17:1 (LPS), phosphatidate 17:0/14:1 (PA), phosphatidylcholine 17:0/14:1 (PC), phosphatidylethanolamine 17:0/14:1 (PE), phosphatidylglycerol 17:0/14:1 (PG), phosphatidylinositol 17:0/14:1 (PI), phosphatidylserine 17:0/14:1 (PS), ergosterol ester 13:0 (EE), triacylglycerol 17:0/17:0/17:0 (TAG), stigmastatrienol, inositolphosphorylceramide 44:0;2 (IPC), mannosyl-inositolphosphorylceramide 44:0;2 (MIPC), mannosyl-di-(inositol-phosphoryl)ceramide 44:0;2 (M(IP)2 C). After extraction, the organic phase was transferred to an infusion plate and dried in a speed vacuum concentrator. The dry extract was resuspended in 7.5 mM ammonium acetate in chloroform/methanol/propanol (1:2:4, V:V:V). The second dry extract was taken up in 33% ethanol solution of methylamine in chloroform/methanol (0.003:5:1; V:V:V). A Hamilton Robotics STARlet robotic platform with the Anti Droplet Control feature for organic solvents pipetting was used for all liquid handling steps. Lipids from CNS, heads or hemolymph homogenates were extracted and analyzed using Lipotype Shotgun Lipidomics technology performed by lipotype (Dresden). Data were analyzed by Lipotype GmbH using their lipid identification software based on LipidXplorer[56,57]. We analyzed three biologically independent samples per genotype. To assess significant differences between the genotypes two-tailed Mann-Whitney rank sum test was used. N indicates the number of independent experiments, while n indicates the number of individual animals. Graphs: means ± standard deviation are shown. Mass spectrometry raw data can be found in supplementary data files 2–4.

### Thin-layer chromatography

Samples were prepared as in[58,59] and lipids extracted by the BUME method[60]. In brief, flies were narcotized with $CO_2$, sorted and dissected on ice. Samples were homogenized with a pestle, lipids extracted with a mixture of butanol:methanol (3:1) at RT and subsequently, organic solvents vaporized using compressed air. Extracted lipids were stored in chloroform/methanol (2:1) solution at −80 °C, HPTLC silicagel 60 plates (Merck) manually loaded with lipid extracts and resolved in chloroform:methanol (1:2). Polar lipids were separated using running buffer PL (chloroform:methanol:water (75:25:2.5)), and neutral lipids were separated using running buffer NL (heptane:diethylether:acetic acid (70:30:1)). Plates were stained with primulin and detected using a Typhoon scanner.

### Quantitative RT-PCR (qPCR)

Oligos used for quantitative RT-PCR are (forward/reverse):
*rp49*: GCTAAGCTGTCGCACAAA/TCCGGTGGGCAGCATGTG
*Pyk*: GCCCACGCTGCCCCATCATC/TCACCAAGACCGGGCTCCTT
*CPT2*: TGGCTTCGGAATCGGCTATT/TTCCCATCGGCCTGGTTTTT
*MTPα*: GCTGTCGGAAGCCATCAGACT/CCACGGGGAAGCCAAACT
*Hmgcl*: TGCACTTACAGCAAAGCG/CGGTCCAACCTCCACAAT
*GLaz*: TGGGACAGATGCCTACGGATT/GCGCTCTACCTCGTACCAGT

Graphs depict mean ± standard deviation.

### Food-choice assay

Yeast food preparation as described[23]. In brief, yeast (BY4741) incubated in lipid-free synthetic medium (SD medium) at 10 or 30 °C were grown until reaching an optical density of $OD_{600} > 5$ (stationary phase). Yeast cultures were pelleted, wet yeast pellets were heat inactivated

and food baits were placed on opposing sides of the assay plates. Subsequently, 12 mated female flies were placed onto assay plates, kept a 29 °C for 12 h and then video-recorded for 3 h at 29 °C. Videos were analyzed manually and feeding events counted (flies feeding on food). Significant differences in numbers of feeding events compared to control animals were determined using one-tailed student's t-test. Graphs: Mean ± standard deviation is shown.

### Brp stainings

Brains from adult mated females were dissected in Graces medium (Thermo Fisher Scientific 11595030) on ice, fixed with 4% PFA in Graces medium at RT for 30 min and probed with anti-Brp (DHB clone NC82) and DAPI (1 mg/ml) overnight at 4 °C in wash solution (5% NGS (normal goat serum), 0.1% TritonX-100, in phosphate-buffered saline, pH=7). Samples were mounted in 50% glycerol. Images of the antennal olfactory lobes were acquired using confocal microscopy (Zeiss LSM 880).

### Statistics & Reproducibility

Description of the statistical analyses used can be found in the figure legends and the respective methods section for each experiment. Data was analyzed double-blind for semithin sections and behavioral assays. For all other data the investigators were not blinded to allocation during experiments and outcome assessment. No statistical method was used to predetermine sample size. For mass spec analysis hemolymph samples that contained cellular contamination were excluded. For the other experiments no data was excluded.

### Reporting summary

Further information on research design is available in the Nature Portfolio Reporting Summary linked to this article.

## Data availability

The data used to generate the graphs shown in this study are provided in the Source Data file. The mass spectrometry and the lifespan data can be found in the supplementary data files. All other data will be made available upon request without undue reservation. Source data are provided with this paper.

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

## Acknowledgements
We thank Cornelia Maas for help with TLCs, Susanne Broschk for help with the LD gradients and Astrid Fleige and Sebastian Goertz for help with semi-thin sections. We thank Lipotype GmbH (Dresden, Germany) for MS analysis. This work was supported by grants from the DFG to M.B. (BR5490/2-1 and FOR2682-TP4) and S.S. (SCHI 1380/2-1).

## Author contributions
H.H., E.M. and K.S. designed and conducted most experiments. Lipi-domic experiments were designed by M.B. and conducted by H.H., E.M. and M.B. L.D. analyzed mitochondrial morphology. SS conceived the study, assisted in designing and interpreting experiments and wrote the manuscript with the help of H.H., E.M., K.S and M.B. S.S. and M.B. obtained funding.

## Funding

## Competing interests
The authors declare no competing interests.
