## [Peer Review File · Nature Communications]

Glycolytically impaired glial cells fuel neural metabolism via β -oxidationThis manuscript has been previously reviewed at another journal that is not operating a transparent peer review scheme. This document only contains reviewer comments and rebuttal letters for versions considered at *Nature Communications*.

REVIEWERS' COMMENTS

Reviewer #1 (Remarks to the Author):

I am satisfied that (for the most part) the authors have done a good job addressing my major concerns where possible. I would only request that they temper their language a bit regarding conclusions. What they claim makes sense, but direct evidence (e.g. from the biochemical assays/in vivo work) is still lacking, and as the other reviewer points out, other roles for key enzymes targeted are known. (I am not concerned that the fly orthologs are doing something different than in mammals--metabolic enzymes are exceedingly well conserved across species.) Pending some softening of the language, this paper will be a nice contribution to the field.

Reviewer #2 (Remarks to the Author):

Both reviewers raised valid concerns with the revised manuscript that the experimental evidence still did not support the claims. They concluded that "the main claims are not supported by direct evidence, rather, they are based on a lot of assumptions, some of which are questionable" (Reviewer 1) and "the revised version still contains major conclusions that are questionable" (Reviewer 2). Many of these major concerns involve the lack of direct measurements for beta-oxidation or ketones.

The latest response letter indicates that many of these overinterpretations have not been corrected, largely because there are no direct measurements for beta-oxidation and ketones. Moreover, the data on the change in mitochondrial morphology (line 76) is interesting but saying that mitochondrial morphology alone is enough to say it is "indicating a glial switch to β -oxidation" seems to be yet another conclusion that may or may not be correct.

Although the evidence does not support the conclusions strongly enough, there is no doubt that this work is in a really interesting area. In addition, I do appreciate that fixing many of the questionable conclusions in the re-revised manuscript satisfactorily would require extensive lipidomics/tracing/metabolomics to measure beta-oxidation and ketones, which as the rebuttal letter says is not trivial, and so could benefit from collaboration. Regarding the inability to measure ketone bodies in fly brains mentioned in the latest response letter. Were acetoacetate, 3-NB or acetone first detected robustly in whole flies (where plenty of material can be obtained)? If so, then the authors statement that they may not have had enough brain material could be valid, otherwise the ketone assays may not be optimal or perhaps these metabolites are just not there in *Drosophila melanogaster* in significant amounts.

Another potential route to publication (not involving substantial additional experiments) is that the paper could be rewritten in a more scientifically precise and rigorous way, that does not make numerous unsubstantiated claims e.g instead of saying (as in abstract) "we show that glycolytically impaired glia switch to fatty acid breakdown via β -oxidation and provide ketone bodies as an alternate neuronal fuel" the wording could be altered to include phrases something like "we show that X and Y genes, predicted to function in beta-oxidation and ketogenesis, are required for...". This would help future proof the study, and the authors, in case counter-evidence were subsequent to come along from lipidomics/tracing/metabolomics analyses that directly measure fatty acid beta-oxidation and ketones in control and knockdown *Drosophila*. Of course this 'toned down' option may no longer support the authors original bold claim of "pioneering evidence of the importance of glial β -oxidation and ketogenesis" (lines 32/33, 55/56) and would necessarily lessen the overall scientific impact of the paper but still leave intact a valuable contribution to the field.

REVIEWERS' COMMENTS

Reviewer #1 (Remarks to the Author):

I am satisfied that (for the most part) the authors have done a good job addressing my major concerns where possible. I would only request that they temper their language a bit regarding conclusions. What they claim makes sense, but direct evidence (e.g. from the biochemical assays/in vivo work) is still lacking, and as the other reviewer points out, other roles for key enzymes targeted are known. (I am not concerned that the fly orthologs are doing something different than in mammals--metabolic enzymes are exceedingly well conserved across species.) Pending some softening of the language, this paper will be a nice contribution to the field.

Thank you for your support. We changed the wording of the manuscript accordingly.

Reviewer #2 (Remarks to the Author):

Both reviewers raised valid concerns with the revised manuscript that the experimental evidence still did not support the claims. They concluded that "the main claims are not supported by direct evidence, rather, they are based on a lot of assumptions, some of which are questionable" (Reviewer 1) and "the revised version still contains major conclusions that are questionable" (Reviewer 2). Many of these major concerns involve the lack of direct measurements for beta-oxidation or ketones.

The latest response letter indicates that many of these overinterpretations have not been corrected, largely because there are no direct measurements for beta-oxidation and ketones. Moreover, the data on the change in mitochondrial morphology (line 76) is interesting but saying that mitochondrial morphology alone is enough to say it is "indicating a glial switch to β -oxidation" seems to be yet another conclusion that may or may not be correct.

We changed the wording.

Although the evidence does not support the conclusions strongly enough, there is no doubt that this work is in a really interesting area. In addition, I do appreciate that fixing many of the questionable conclusions in the re-revised manuscript satisfactorily would require extensive lipidomics/tracing/metabolomics to measure beta-oxidation and ketones, which as the rebuttal letter says is not trivial, and so could benefit from collaboration. Regarding the inability to measure ketone bodies in fly brains mentioned in the latest response letter. Were acetoacetate, 3-NB or acetone first detected robustly in whole flies (where plenty of material can be obtained)? If so, then the authors statement that they may not have had enough brain material could be valid, otherwise the ketone assays may not be optimal or perhaps these metabolites are just not there in *Drosophila melanogaster* in significant amounts.

\$\beta\$ -hydroxybutyrate was successfully measured in whole animals (Luong et al. (2006). *Cell Metabolism*, 4(2), 133–142. <https://doi.org/https://doi.org/10.1016/j.cmet.2006.05.013>). Thus, *Drosophila* has been shown to produce ketone bodies. We were not able to find any publications reporting tissue-specific measurements. As the neurons presumably metabolize the \$\beta\$ HB that is produce in the brain, we do not expect a high build-up of \$\beta\$ HB, and thus we assume that we were not able to obtain enough material to reliably measure \$\beta\$ HB in brain samples with the method used here.

Another potential route to publication (not involving substantial additional experiments) is that the paper could be rewritten in a more scientifically precise and rigorous way, that does not make numerous unsubstantiated claims e.g instead of saying (as in abstract) "we show that glycolytically

impaired glia switch to fatty acid breakdown via β -oxidation and provide ketone bodies as an alternate neuronal fuel" the wording could be altered to include phrases something like "we show that X and Y genes, predicted to function in beta-oxidation and ketogenesis, are required for...". This would help future proof the study, and the authors, in case counter-evidence were subsequent to come along from lipidomics/tracing/metabolomics analyses that directly measure fatty acid beta-oxidation and ketones in control and knockdown *Drosophila*. Of course this 'toned down' option may no longer support the authors original bold claim of "pioneering evidence of the importance of glial β -oxidation and ketogenesis" (lines 32/33, 55/56) and would necessarily lessen the overall scientific impact of the paper but still leave intact a valuable contribution to the field.

We changed the wording of the manuscript accordingly.